# Sex-specific processing of social cues in the medial amygdala

Joseph F Bergan[1†], Yoram Ben-Shaul[2†], Catherine Dulac[1*]

[1]Molecular and Cellular Biology, Howard Hughes Medical Institute, Harvard University, Cambridge, United States; [2]School of Medicine, Department of Medical Neurobiology, The Hebrew University of Jerusalem, Jerusalem, Israel

**Abstract** Animal–animal recognition within, and across species, is essential for predator avoidance and social interactions. Despite its essential role in orchestrating responses to animal cues, basic principles of information processing by the vomeronasal system are still unknown. The medial amygdala (MeA) occupies a central position in the vomeronasal pathway, upstream of hypothalamic centers dedicated to defensive and social responses. We have characterized sensory responses in the mouse MeA and uncovered emergent properties that shed new light onto the transformation of vomeronasal information into sex- and species-specific responses. In particular, we show that the MeA displays a degree of stimulus selectivity and a striking sexually dimorphic sensory representation that are not observed in the upstream relay of the accessory olfactory bulb (AOB). Furthermore, our results demonstrate that the development of sexually dimorphic circuits in the MeA requires steroid signaling near the time of puberty to organize the functional representation of sensory stimuli.

*For correspondence: dulac@
fas.harvard.edu

†These authors contributed
equally to this work

Competing interests: See
page 18

Reviewing editor: Peggy Mason,
University of Chicago, United
States

## Introduction

Throughout the animal kingdom, dedicated signals allow individuals to identify and distinguish members of their own species: songbirds attract mates during bouts of singing, cichlid fish communicate dominance status with dramatic changes in body coloration, and rodents signal their social status through the emission of chemical cues (*Brainard and Doupe, 2002*; *Tinbergen, 1951*; *Maruska and Fernald, 2011*; *Dulac and Kimchi, 2008*). In turn, the execution of instinctive behaviors such as mating, parenting, territorial defense, and predator avoidance is triggered and modulated by sensory signals within a given physiological context, including the sex, endocrine, or developmental status of an individual (*Tinbergen, 1951*; *Insel and Fernald, 2004*). However, the brain circuits that organize species- and sex-specific instinctive behaviors in response to animal cues are still poorly characterized.

Chemosensory communication in rodents offers a unique opportunity to explore the organizational and functional principles of social behavior circuits. Impairment in either vomeronasal or olfactory signaling results in dramatic deficits in predator avoidance and social communication (*Bean and Wysocki, 1989*; *Stowers et al., 2002*; *Mandiyan et al., 2005*; *Yoon et al., 2005*; *Kimchi et al., 2007*; *Kobayakawa et al., 2007*; *Isogai et al., 2011*), demonstrating that chemical cues are essential mediators of animal–animal recognition in mice and that the main olfactory system and the vomeronasal pathway work in concert to guide behavioral responses to chemosensory input (*Dulac and Wagner, 2006*; *Dulac and Kimchi 2008*). The main olfactory epithelium (MOE) and vomeronasal organ (VNO) are distinguished by the types of chemical cues that each detects: volatile compounds are primarily detected by the MOE while non-volatile signals are preferentially processed by the VNO. At the circuit level, neurons in the MOE project to the main olfactory bulb (MOB), which in turn projects axons to the lateral amygdala, primary olfactory cortex, and ultimately to areas involved in the cognitive processing of odorants (*Sosulski et al., 2011*). In addition, neurons expressing the neuropeptide

**eLife digest** Many animals emit and detect chemicals known as pheromones to communicate with other members of their own species. Animals also rely on chemical signals from other species to warn them, for example, that a predator is nearby. Many of these chemical signals—which are present in sweat, tears, urine, and saliva—are detected by a structure called the vomeronasal organ, which is located at the base of the nasal cavity.

When this organ detects a particular chemical signal, it broadcasts this information to a network of brain regions that generates an appropriate behavioral response. Two structures within this network, the accessory olfactory bulb and the medial amygdala, play an important role in modifying this signal before it reaches its final destination—a region of the brain called the hypothalamus. Activation of the hypothalamus by the signal triggers changes in the animal's behavior. Although the anatomical details of this pathway have been widely studied, it is not clear how information is actually transmitted along it.

Now, Bergan et al. have provided insights into this process by recording signals in the brains of anesthetized mice exposed to specific stimuli. Whereas neurons in the accessory olfactory bulb responded similarly in male and female mice, those in the medial amygdala showed a preference for female urine in male mice, and a preference for male urine in the case of females. This is the first direct demonstration of differences in sensory processing in the brains of male and female mammals.

These differences are thought to result from the actions of sex hormones, particularly estrogen, on brain circuits during development. Consistent with this, neurons in the medial amygdala of male mice with reduced levels of estrogen showed a reduced preference for female urine compared to control males. Similarly, female mice that had been previously exposed to high levels of estrogen as pups showed a reduced preference for male urine compared to controls.

In addition to increasing understanding of how chemical signals—including pheromones—influence the responses of rodents to other animals, the work of Bergan et al. has provided clues to the neural mechanisms that underlie sex-specific differences in behaviors.

GnRH, a master regulator of reproduction in vertebrates, receive major inputs from primary olfactory areas (*Boehm et al., 2005*; *Yoon et al., 2005*). In contrast, vomeronasal circuits largely bypass cognitive centers, as sensory neurons of the VNO send axons to the accessory olfactory bulb (AOB), which in turn projects to nuclei of the medial amygdala (MeA), the bed nucleus of the stria terminalis, and the hypothalamus (*Petrovich et al., 2001*), areas involved in controlling innate behaviors and neuroendocrine changes.

Recent studies have begun to explore the basic principles of vomeronasal information coding that lead to specific social and defensive behavioral responses. The identification of key components of VNO signal transduction, such as the TRPC2 ion channel (*Liman et al., 1999*), and the members of the V1R and V2R families of vomeronasal receptors (VRs) (*Dulac and Axel, 1995*; *Herrada and Dulac, 1997*; *Matsunami and Buck, 1997*; *Ryba and Tirindelli, 1997*; *Dulac and Torello, 2003*), has established the molecular foundation of vomeronasal sensing. In vitro recording and imaging of VNO neuronal activity after exposure to various stimuli, as well as in vivo characterization of the vomeronasal receptor response profile to a wide range of animal cues, have revealed that distinct populations of VNO receptors identify ethologically relevant chemosignals with high specificity and that signal detection in the VNO plays an essential role in distinguishing behaviorally relevant sensory information (*Holy et al., 2000*; *Leinders-Zufall et al., 2000*; *Nodari et al., 2008*; *Leinders-Zufall et al., 2009*; *Haga et al., 2010*; *Isogai et al., 2011*; *Turaga and Holy, 2012*).

In contrast to the apparent specificity of VNO responses for distinct animal cues, a large proportion of output neurons in the AOB, the primary targets of VNO sensory projections, were shown to respond to multiple classes of stimuli (*Luo et al., 2003*; *Hendrickson et al., 2008*; *Ben-Shaul et al., 2010*). In particular, many AOB mitral/tufted cells respond with similar strength to cues such as female and predator odors, which are clearly associated with opposing behavioral outcomes (*Ben-Shaul et al., 2010*). The complex responses of AOB mitral/tufted cells, together with their distinctive multi-branched dendritic trees, suggest that they integrate sensory information across distinct vomeronasal receptors

(*Wagner et al., 2006*). Ultimately, downstream vomeronasal centers that control behavioral outputs must be able to interpret and disambiguate the relevant information from these broad patterns of AOB activity.

Several lines of evidence suggest that the MeA plays a central role in the vomeronasal–sensorimotor transformation that leads to specific behavioral responses. The MeA is the primary recipient of AOB inputs, it projects to distinct nuclei of the hypothalamus involved in social and defensive responses (*Petrovich et al., 2001*; *Choi et al., 2005*; *Lin et al., 2011*) and disruptions in MeA signaling cause profound deficits in social and predator recognition (*Ferguson et al., 2001*; *Li et al., 2004*). However, despite this central position in the vomeronasal sensory pathway, little is known about how neuronal activity in the MeA participates in the transformation of AOB inputs into behaviorally relevant signals.

We have investigated the characteristics of MeA neuronal activity in both male and female mice in response to conspecific and heterospecific chemosignals. Our results uncovered several emergent features in the MeA representation of olfactory information that are not present in either VNO or AOB responses. These features suggest how key sensory parameters relevant for sex-specific, social and defensive behaviors are extracted from the complex activity pattern of the AOB. Our findings provide significant insights into the neural processing of social cues and open new avenues for further understanding the emergence of sex-specific behaviors.

## Results

### Extracellular recordings in the MeA of anesthetized animals

To investigate how the MeA responds to chemosensory cues, we combined multisite extracellular recordings with sensory stimulation of the VNO in anesthetized mice (*Figure 1A*). Linear electrode arrays (Neuronexus) were positioned dorsal to the MeA based on stereotaxic coordinates (1.7-2.0 mm lateral and caudal from bregma; *Franklin and Paxinos, 2007*) and advanced until the electrode tip reached the ventral brain surface. This targeting strategy was intentionally chosen to focus on the posterior MeA (MeApd and MeApv), allowing simultaneous sampling of single and multi-unit activity from 32 evenly spaced sites distributed between MeA subnuclei with distinct functions. MeA units responsive to sensory stimuli were localized to the ventral surface of the brain and extended approximately 1 mm dorsally, consistent with the anatomy of the posterior MeA. The locations of all recording sites were confirmed through postmortem histology of dye-labeled electrode tracks (*Figure 1B*).

Single units were distinguished from multi-unit activity based on spike shape, separation of principal component projections, and the presence or absence of a refractory period between successive spikes (*Figure 1—figure supplement 1*; *Harris et al., 2000*; *Hazan et al., 2006*). In most experiments, ~2–6 recording sites were located dorsal to the MeA and were easily distinguished from the units in the MeA, which remain quiet in the absence of vomeronasal stimulation. Multiple MeA units were recorded simultaneously during each recording session (17.5 ± 11.7 units/session) with ~40% (6.6 ± 6.4 units/session) identified as single units. In this study, 197 units (82 single units; 115 multi units) recorded from the MeA of 55 adult BALB/c mice (10–12 weeks old; sexually naive) responded to sensory stimuli (p<0.01, non-parametric ANOVA).

Baseline firing rates of single MeA units were typically very low (0.21 ± 0.02 Hz; *Figure 1—figure supplement 2*). Our relatively unbiased multi-electrode sampling approach allowed identification of many neurons that did not display any spiking activity prior to stimulus presentation. Upon VNO stimulation, MeA units responded in stimulus-locked and stimulus-specific manner (*Figure 1C*) that resembled sensory responses observed in the AOB in both latency and duration (*Ben-Shaul et al., 2010*). MeA units typically increased their firing rate after a preferred stimulus was presented reaching a peak firing rate (8.1 ± 0.31 Hz) ~5 s after stimulus presentation. Units with significant responses were identified by comparing the spike rates prior to stimulus presentation to spike rates following stimulus presentation using a non-parametric ANOVA performed at the significance level of p≤0.01 (see 'Materials and methods').

### Olfactory and VNO-evoked responses in the MeA

The MeA has been described as receiving main olfactory projections in addition to vomeronasal input (*Kang et al., 2009*; *Martinez-Marcos, 2009*). This prompted us to investigate the respective contributions of MOE and VNO stimulations to MeA responses. MOE stimulation elicited clear

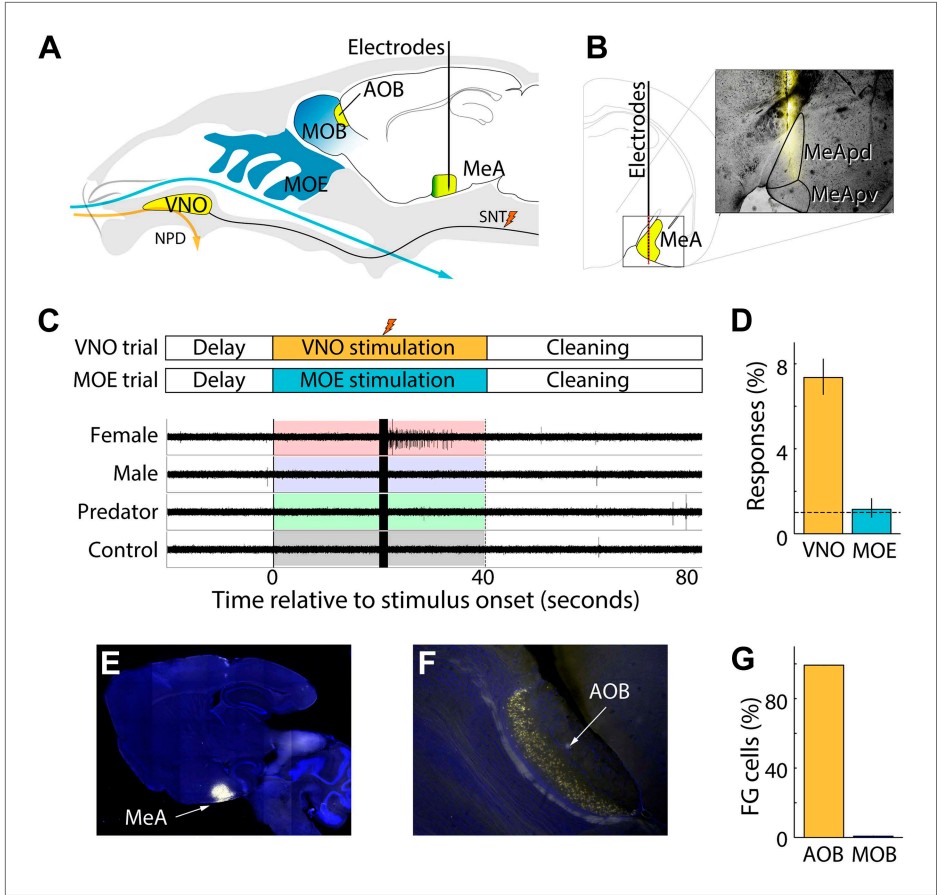

**Figure 1**. Experimental system for recording MeA sensory responses. (**A**) Vomeronasal and olfactory structures are shown in yellow and blue, respectively. Multichannel electrophysiological probes are stereotaxically positioned in the MeA to continuously record neural responses to sensory stimulation. VNO stimulus presentation (orange arrow) is achieved by placing nonvolatile stimuli in the nostril followed by electrical stimulation of the sympathetic nerve trunk (SNT) to activate the VNO pump and permit access of stimuli into the VNO. VNO stimuli are washed out through the NPD. MOE stimulation is achieved by controlling airflow of volatile stimuli into the nostril (blue arrow), which access the MOE, and are eliminated through a tracheotomy. (**B**) Diagram illustrating a coronal section through the posterior MeA with red dots indicating the expected dorsal–ventral distribution of recording sites. A single fluorescent electrode tract accurately targeted to MeA is shown in the inset. (**C**) Timecourse of VNO and MOE stimulation trials (top). Electrophysiological signals recorded from a single MeA electrode during four successive trials reveal a well-isolated unit responding only to female stimuli following electrical stimulation of SNT (stimulation artifacts are evident at 20 s). (**D**) Percentage of MeA units responding to VNO vs MOE stimulation with chance rates indicated by a horizontal dashed line. (**E**) Sagittal section of whole brain showing DAPI staining and the site of MeA FluoroGold iontophoresis. (**F**) Dense retrograde labeling of AOB projection neurons. (**G**) Fraction of AOB (99.8%) and MOB (0.2%) neurons that are retrogradely labeled by FluoroGold iontophoresis in the MeA.

The following figure supplements are available for figure 1:

**Figure supplement 1**. Clustering and analysis of multichannel electrophysiological recordings.

**Figure supplement 2**. Baseline electrophysiological characteristics of MeA responses.

**Figure supplement 3**. MOE-driven responses in the MOB and PLCO.

olfactory-evoked neuronal activity in the MOB and the posterolateral cortical amygdala (PLCO) in response to volatile urinary cues and odorants (***Figure 1—figure supplement 3***). However, while we observed robust VNO-mediated responses in the MeA, we did not detect stimulus-specific responses for MOE-delivered stimuli (units responsive to MOE stimuli at $p \leq 0.01$: 1.1%, 95% CI: 0.8–1.6%; ***Figure 1D***).

Rather, in agreement with *Samuelsen and Meredith (2009)* and *Miyamichi et al. (2011)*, we found little evidence that distinct volatile stimuli elicit specific sensory responses in the MeA. Therefore, the vomeronasal system emerged as the dominant sensory input to the MeA units we recorded.

We further investigated MOB to MeA connectivity by iontophoretically injecting FluoroGold into the MeA at the same stereotaxic location used for electrophysiology. All injection sites (6 animals) encompassed most of the posterior MeA, with more diffuse labeling extending both anterior and posterior (*Figure 1E*). Rare neurons were found retrogradely labeled in the posterior MOB, consistent with previous reports (*Kang et al., 2011*). However, we found that FluoroGold retrogradely labeled ~100 times more AOB than MOB neurons, clearly demonstrating the dominance of AOB inputs, especially in light of the much larger number of MOB compared to AOB neurons (*Figure 1F,G*; see *Miyamichi et al., 2011*). Therefore, functional and anatomical evidence confirms that the MeA, as targeted in these experiments, receives input predominantly from the AOB (*Pro-sistiaga et al., 2007*). Modulatory influences are important for a variety of sensory pathways (*King and Palmer, 1985*; *Sherman and Guillery, 1998*; *Bergan and Knudsen, 2009*) and any main olfactory inputs to the MeA cells recorded here are likely subtler than our experiments were designed to detect.

## Stimulus-evoked single neuron activity in the MeA

MeA stimuli-evoked responses typically followed electrical stimulation of the sympathetic nerve trunk, which induces uptake of stimuli into the VNO lumen. Some evoked responses appeared after stimulus application but before stimulation of the sympathetic nerve trunk suggesting that stimuli can gain access to the VNO lumen in our anesthetized preparation, likely due to occasional spontaneous VNO pump activation. In these cases, aside from the onset time of response, response properties of individual units were indistinguishable from responses following nerve trunk stimulation. Based on these observations, our analysis includes the full epoch in which sensory stimuli activate the VNO.

Vomeronasal sensory neurons detect diverse non-volatile chemical cues found in the urine, tears, saliva, and sweat of other animals (*Lin da et al., 2006*; *He et al., 2008*; *Haga et al., 2010*). While all of these stimulus sources are likely to convey important sensory information about the sex, age, and behavioral state of other animals, we initially focused on urine stimuli from predators, female mice, and male mice as these readily available stimuli elicited vigorous responses from single MeA units. Unless otherwise noted, predator stimuli were a mixture of bobcat, fox, and rat urine diluted 1:100 in Ringer's solution; female and male stimuli were a mixture of urine from BALB/c, C57, and CBA strains diluted 1:100 in Ringer's solution. Sensory responses of MeA units typically exhibited high specificity for one stimulus or a subset of the tested stimuli (*Figure 2*), with a smaller number of MeA units responding broadly to multiple stimuli from both conspecific and non-conspecific sources (*Figure 2*, bottom four rows).

One of the best-known functional features of the MeA, demonstrated via immediate early gene studies, is the segregation into dorsal and ventral processing subdivisions. The posterior dorsal MeA, or MeApd, is essential for reproductive and social behaviors while the ventral MeA, or MeApv, is essential for defensive behaviors (*Fernandez-Fewell and Meredith, 1994*; *Choi et al., 2005*; *Wu et al., 2009*). We stereotaxically targeted the posterior MeA such that the array of recording sites would span both the MeApv and MeApd. Consistent with previous findings, we found that a clear topographical segregation of sensory responses is apparent in the real time response properties of MeA units. Neurons that responded most strongly to 'conspecific' stimuli were typically located dorsal to neurons that responded most strongly to 'predator' stimuli, and this was true for data recorded from male or female animals (*Figure 2—figure supplement 1*). However, we also observed MeA neurons that responded significantly to multiple stimuli with varying response strengths (*Figure 2—figure supplement 1A*), indicating that individual neurons distributed throughout the MeA have access to information about both conspecific and defensive stimuli.

## Sharpened stimulus representations in the MeA

Units recorded in the AOB, the sole target of VNO output and the primary input to the MeA, display a wide range of response selectivity with some units responding similarly to seemingly contradictory sensory signals (*Hendrickson et al., 2008*; *Ben-Shaul et al., 2010*). This finding prompted us to compare the level of specificity of sensory responses in the MeA and AOB.

The AOB external cellular layer was targeted by making a small craniotomy immediately rostral to the rhinal sinus and advancing electrodes into the AOB at a 30° angle. Consistent with all previous

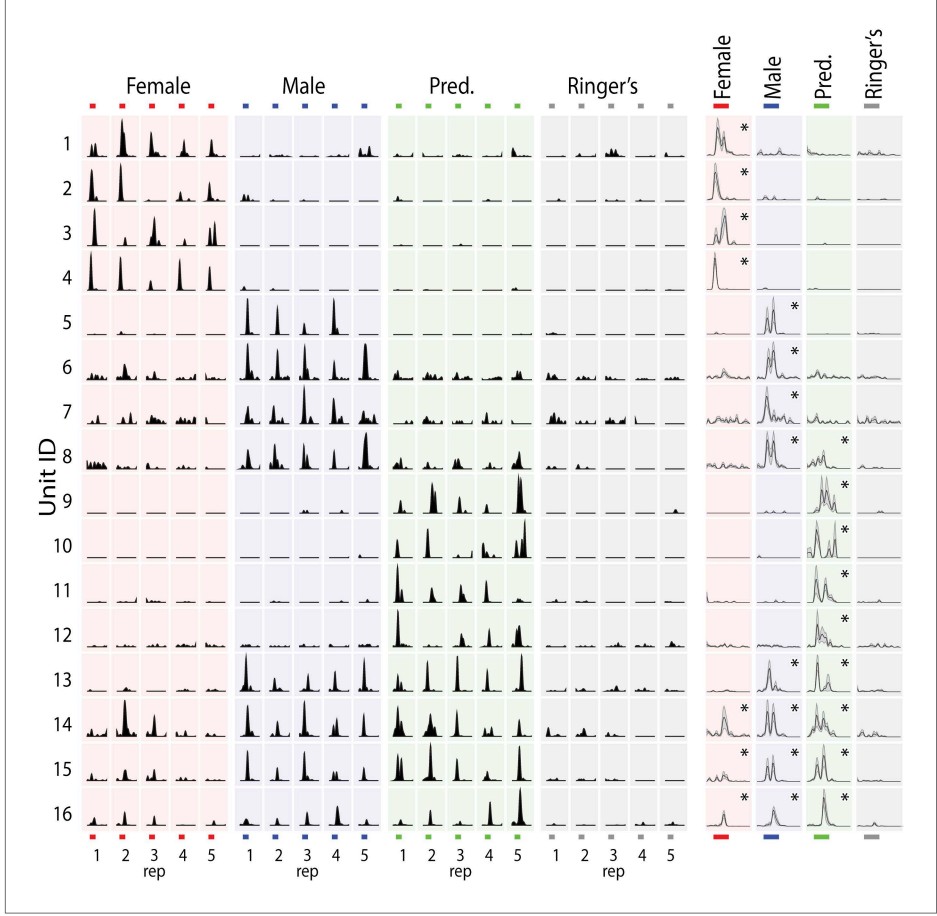

**Figure 2**. MeA sensory responses to VNO stimuli. (**Left**) Each row shows the responses elicited in a single MeA unit by four different VNO stimuli, with each stimulus presented 5 times. The order of stimulus presentation was randomized during the experiment, but is shown grouped by stimulus for clarity. (**Right**) Histograms showing the mean response and standard error (shaded region) for each unit. Responses were aligned to the onset of stimulus presentation. All significant responses (p<0.01; Nonparametric ANOVA) are indicated by an asterisk in the top right corner of the average histogram plots. Response magnitudes for each unit were normalized to the maximum response for all stimuli. Colored bars (top and bottom) indicate the 40 s epoch following stimulus presentation that was considered for all analyses.

The following figure supplements are available for figure 2:

**Figure supplement 1**. MeA units responsive to different stimulus categories are spatially segregated.

**Figure supplement 2**. AOB sensory responses to VNO stimuli.

electrophysiology in the MOB (*Bhalla and Bower, 1997*; *Kay and Laurent, 1999*; *Rinberg et al., 2006*; *Doucette et al., 2011*) and AOB (*Ben-Shaul et al., 2010*), showing that granule cells are virtually undetectable through extracellular recordings, units responsive to VNO stimuli were only detected from electrodes located in the mitral cell layer. Similarly, systematic post-mortem histology from experiments yielding isolated units invariably confirmed accurate targeting to the AOB mitral cell layer (*Figure 2—figure supplement 2*). Thus, we can assume with a high level of confidence that we chiefly recorded from AOB projection neurons and not interneurons.

Male mouse, female mouse, and predator stimuli each elicited robust responses in the MeA and AOB. Using a threshold for statistical significance of p<0.01, the fractions of units showing sensory responses are 6.3, 5.1, and 6.9% for female, male, and predator stimuli respectively for MeA units, as compared to 16.6, 11.4, and 21.3% for AOB units (p<0.01, non-parametric ANOVA). Thus, the

percentage of units responding to each stimulus is dramatically lower in the MeA than it is in the AOB (*Figure 3A*). Given the extremely low activity of MeA units in the absence of sensory stimuli, even small responses can reach high statistical significance (see *Figure 2*: units 8 and 15). To further compare the stimulus selectivity of individual neurons, we calculated a selectivity index that ranges linearly from 0 (all stimuli elicit equal responses: center of triangles in panels C, D) to 1 (only one stimulus elicits a response: vertices of triangles in panels C, D) for units recorded from the AOB (mean 0.38; 95% CI: 0.36 to 0.41; see Supplementary Information) and from the MeA (mean 0.54; 95% CI: 0.50 to 0.58). Comparison of the response distributions between the AOB and the MeA reveals a clear and significant shift to values with higher selectivity in the MeA (*Figure 3B*; p<0.0001; permutation test).

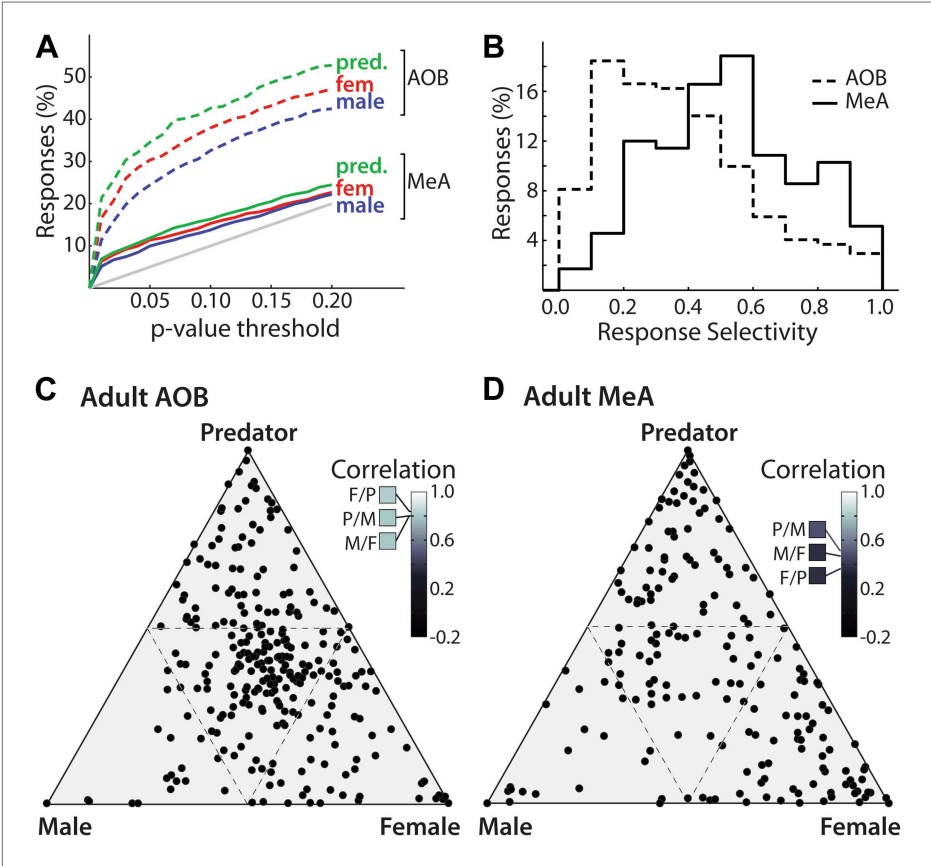

**Figure 3**. Decreased frequency and increased specificity of sensory responses in the MeA compared to the AOB. (**A**) The percentage of single AOB units (dashed curves) and MeA units (solid curves) exhibiting statistically significant responses to male (blue), female (red), and predator (green) vomeronasal stimuli as the threshold for inclusion was varied from p<0 to p<0.2 (abscissa). The diagonal gray line indicates the predicted false positive rate. (**B**) Distribution of response selectivity ('Materials and methods') showing a shift towards higher specificity in MeA (solid line) as compared to the AOB (dashed line). (**C**) Selectivity of sensory responses for units recorded in the adult AOB (197 units from male and female animals). (**D**) Selectivity of sensory responses for units recorded in the adult MeA (274 units from male and female animals). Each point represents the response profile of an individual unit, with at least one significant response, to male, predator, and/or female stimuli. Points located near a vertex (more frequent in the MeA) represent units that respond most strongly for the stimulus indicated at that vertex whereas points at the center (more frequent in the AOB) represent units that respond similarly to all stimuli. Insets (**C** and **D**) show correlation between responses for each pair of stimuli.

The following figure supplements are available for figure 3:

**Figure supplement 1**. Leverage analysis for stimulus response correlations.

**Figure supplement 2**. Principal components analysis of MeA categorization data.

The emergence of selectivity in the MeA is further shown by plotting the responses of all units on a triangular axis for which the position of each point indicates the relative magnitude of the responses elicited by each of three stimuli (*Figure 3C*: AOB, *Figure 3D*: MeA). The vertices represent exclusive responses to one of the stimuli. Comparison of the spatial distribution of individual neurons between the AOB and the MeA indicates that, whereas AOB units tend to populate the central region of the triangular plot, MeA units more evenly span the entire area with an increased density at the vertices. Dividing the triangular response space into four equal quadrants (dashed lines *Figure 3C,D*) showed that 44.4% of AOB units vs 21.6% of MeA units reside in the central quadrant associated with less specific responses. Thus, flow of information from the AOB to the MeA is associated with a sharpening of stimulus-driven responses. The increased selectivity of sensory responses in the MeA is similarly reflected by a marked decorrelation of responses evoked by different sensory stimuli across the MeA population (AOB: *Figure 3C* inset; MeA: *Figure 3D* inset; *Figure 3—figure supplement 1*).

The sharpened response pattern observed in the MeA (*Figure 3*) suggests that MeA activity could represent higher-level categorical information. We measured responses in male mice to stimuli from three ethologically defined classes: female (estrus: BALB/c, CBA, and C57), male (BALB/c, CBA, and C57), and defensive cues (fox, bobcat, and rat urine). A principal component analysis performed on responses evoked by these stimuli revealed that the first PC reflected the distinction between 'female' and 'predator' categories and accounted for 41% of the total response variance (*Figure 3—figure supplement 2*). The second PC distinguished between 'female' vs 'male' stimuli and accounted for 19% of the total variance. Therefore, most of the variation of sensory responses in the MeA of male mice is devoted to the distinction between 'female' from 'predator' and 'male' categories respectively. However, individual sensory responses in the MeA rarely form sharply defined stimulus categories, but rather, maintain enough information to distinguish between individual elements of each category (*Figure 3—figure supplement 2*).

## Sexually dimorphic responses in the MeA

Sexually dimorphic behaviors of male and female animals to conspecific stimuli represent a dramatic and reproducible example of individual variability: male stimuli elicit territorial responses from another male mouse, but reproductive displays from a receptive female mouse. Sex-specific behaviors imply the existence of sexually dimorphic sensory processing; however, little is known about in vivo functional differences between the sexes at the single neuron level.

To assess the role of the MeA for sex-specific computation within the vomeronasal pathway, we compared the activity of MeA units recorded from male vs female animals in response to urinary stimuli from male and female conspecifics (*Figure 4*). This analysis revealed a striking sexual dimorphism in the stimulus selectivity of MeA units (*Figure 4B*), such that MeA responses are dramatically more frequent to opposite-sex stimuli. We quantified the sex preference of individual neurons using an index that ranges from −1 to 1, reflecting responses exclusive to male or female stimuli, respectively. Distributions of responses from both male and female MeA neurons appear strikingly skewed away from zero, toward stimuli representing the opposite sex (*Figure 4E*; p<0.0001 permutation test). Indeed, 82.7% of male MeA units respond more strongly to female stimuli and 83.3% of female MeA units respond stronger to male stimuli. Thus, for both sexes, there is an unmistakable overrepresentation of responses to stimuli from the opposite sex. In addition, the ratio of predator to conspecific responsive units was significantly higher in the MeA of female vs male mice (*Figure 4—figure supplement 1*).

The sexually dimorphic responses observed in the MeA could be generated by specific neuronal processing within the MeA or inherited from sexually dimorphic response patterns at the AOB. We compared the responses of AOB units recorded in both male and female mice, and, in contrast to the MeA sensory representation, found little evidence for sexual dimorphism (*Figure 4A,D*). Rather, the responses of AOB units recorded from both male and female animals share a moderate bias in responses for female stimuli (*Figure 4D*). Thus, the profound sexual dimorphism of sensory representations described here is an emergent property of the MeA.

## Development of MeA sexual dimorphism

Circuits underlying sexually dimorphic behaviors are organized perinatally by the early action of steroid hormones, and activated when an animal reaches reproductive maturity. This developmental schema, known as the organizational/activational hypothesis of sexual dimorphism in the brain (*Phoenix et al., 1959*; *McCarthy, 2008*), proposes the establishment of sex-specific circuit features

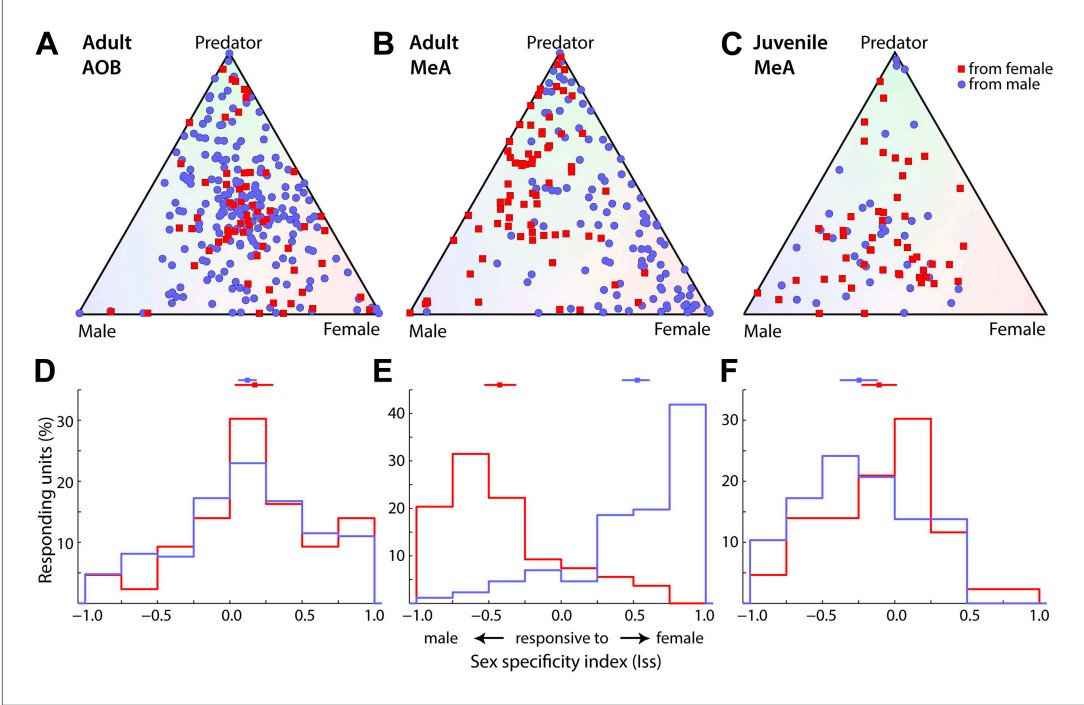

**Figure 4**. Sexual dimorphism of adult MeA responses. (**A**) Responses of AOB neurons to vomeronasal stimuli in adult male (210 units) and female (64 units) mice. (**B**) Responses of MeA neurons to vomeronasal stimuli in adult male (106 units) and female (91 units) mice. (**C**) Responses of MeA neurons to vomeronasal stimuli in juvenile male (37 units) and female (50 units) mice. Units shown in panels **A**–**C** are the same data shown in *Figure 3C,D* but classified according to the sex of the animal recorded. Blue circles indicate units recorded from male mice and red squares indicate data recorded from female mice. (**D**–**F**) Sex-specificity ('Materials and methods') histograms are shown for all units recorded from male (blue) and female (red) animals in the adult AOB (**D**), adult MeA (**E**) and juvenile MeA (**F**). Horizontal lines (above) indicate the mean and 95% confidence interval (bootstrap CI) of the mean for each distribution. Data collected from males vs females were only different in the adult MeA (AOB: p=0.26 adult MeA: p<0.00001; juvenile MeA: p=0.18; permutation tests).

The following figure supplements are available for figure 4:

**Figure supplement 1**. Sexual dimorphism in the dominance of predator versus conspecific responses.

---

during a perinatal 'organizational' stage that are subsequently made functional during a peripubertal 'activational' stage. To understand when the MeA circuitry acquires its sex-specific functional features, we recorded responses to sensory stimuli in the MeA of juvenile male and female mice at 18–21 days, which is roughly 1 week prior to puberty onset (28 days; *Nathan et al., 2006*). We found that, while single units recorded from the MeA of juvenile mice responded to VNO stimuli, they were far less selective for male, female, and predator cues (*Figure 4C,F*) than those in the adult MeA (*Figure 4B,E*), and instead showed closer similarity with the response selectivity identified in the adult AOB (*Figure 4A*). Thus, 34.7% of juvenile units were in the central quadrant associated with less specific responses, as opposed to 21.6% in the adult MeA. The selectivity of MeA responses to opposite sex cues is absent in juvenile animals, and was not significantly different between males and females (*Figure 4F*; p=0.06, permutation test), a feature also shared with the adult AOB and strikingly different from the adult MeA. Thus, MeA responses in juvenile mice are not sexually dimorphic, even though the perinatal organizational phase of MeA sexual development, during which steroid-dependent cell death generates sexually dimorphic MeA cell numbers, is essentially complete (*Wu et al., 2009*), and appear instead highly reminiscent to the responses observed in the AOB.

## Hormone dependence of MeA sexual dimorphism

Much of testosterone's influence on sexually dimorphic patterning of the brain is achieved after its conversion to estrogen by aromatase and activation of estrogen receptors. Therefore, aromatase

expression in the brain is a key factor in the development of sexually dimorphic behaviors and brain circuits. Although sparse in the rodent brain, the MeApd is a primary site of aromatase expression, and sexual dimorphisms in MeA regional volume, cell number, and neural connectivity are mediated by aromatase-dependent sex-steroid signaling (see *Cooke et al., 1999*; *McCarthy, 2008*).

The dorsal bias in aromatase expression parallels a dorsal bias of sexually dimorphic sensory responses we observed. We therefore reasoned that the sexually dimorphic functional differences observed here may be impaired in mice lacking aromatase function, and investigated sensory responses in the MeA of adult aromatase knockout mice (ArKO$^{-/-}$; *Fisher et al., 1998*). As predicted, MeA units from ArKO$^{-/-}$ males frequently responded to multiple stimuli; whereas, MeA units from age matched heterozygous littermates (ArKO$^{+/-}$) were similar to those observed in wild-type males (*Figure 5A*). Indeed, the sex specificity of neurons recorded in adult ArKO$^{-/-}$ males is largely symmetric around 0, and presents a pattern of responses intermediate to those observed in intact juvenile and adult male mice (*Figure 5B*). In contrast, adult heterozygous littermates (ArKO$^{+/-}$) displayed a bias towards responses to female stimuli similar to that observed in wild-type males (*Figure 5A,B*). Like juvenile animals, most (42.1%) of MeA units recorded from adult ArKO$^{-/-}$ males reside in the central quadrant associated with less specific responses, compared to 19.5% of units recorded from ArKO$^{+/-}$. Similarly, response patterns to different stimuli are much more correlated in the MeA of ArKO$^{-/-}$ males (*Figure 5C*, bottom) than in ArKO$^{+/-}$ littermates (*Figure 5C*, top) or wild-type males (*Figure 3D*), reflecting a loss of discriminability at the individual unit level. These results suggest that estrogen signaling (via conversion of testosterone by aromatase) is essential for generating the pattern of sensory responses observed in the adult male MeA. The fact that adult ArKO$^{-/-}$ males are also different from juvenile males indicates that other processes, such as direct testosterone signaling, may play an important role in developing the adult MeA sensory representation.

To determine if perinatal hormone exposure is sufficient to establish a masculine pattern of MeA neural function, we injected newborn female pups with estradiol–benzoate at three postnatal days (p0, p7, and p14), which has been shown to induce a male-like aromatase expression pattern and a subset of male behaviors in adult female mice (*MacLusky and Naftolin, 1981*; *Wu et al., 2009*). We then recorded sensory evoked MeA responses in these females when they reached adulthood (10–14 weeks).

Neurons responding most strongly to same-sex stimuli are nearly absent from the MeA of untreated adult females. However, in estrogen treated females, 51.2% of MeA neurons respond more strongly to female than to male stimuli, with 26.7% responding to female stimuli at least 3 times stronger than to male stimuli (*Figure 6A,B*). This novel subset of neurons (female responsive in female animals) is located more dorsally relative to predator-responsive neurons, as is typical of female responsive neurons in males (*Figure 6C*). In addition, estrogen treatment resulted in a clear reduction of the number of neurons selective for male stimuli (*Figure 6B*).

## Discussion

Our findings identify the MeA as a central node of the vomeronasal sensory–motor transformation,

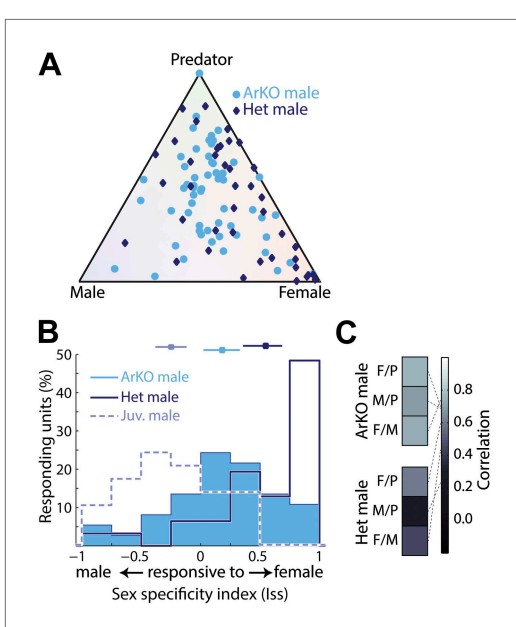

**Figure 5**. Importance of aromatase signaling for the development of sexually dimorphic MeA responses. (**A**) Population summary of MeA responses to male, female, and predator stimuli recorded from adult male ArKO mice (light blue circles) or heterozygous male littermates (dark blue diamonds). All plotted units responded significantly to at least one stimulus. (**B**) Sex-specificity histograms for units recorded from ArKO males (blue fill), heterozygous male littermates (dark blue; no fill), and wild-type juvenile males (light blue; no fill). Horizontal lines indicate the mean and 95% confidence interval of the mean of each distribution. (**C**) Matrices of correlation for the population responses between pairs of sensory stimuli for heterozygous males (top) and ArKO males (bottom).

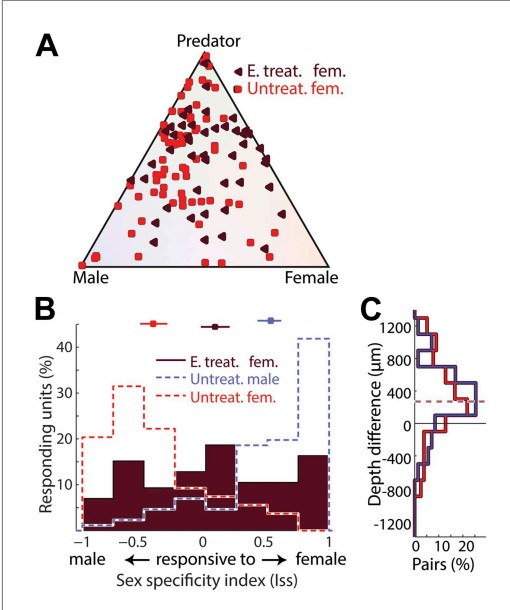

**Figure 6**. Estrogen influences the development of sexually dimorphic MeA responses. (**A**) Population summary of MeA responses to male, female, and predator stimuli recorded from adult untreated female mice (red squares) or estrogen treated adult females (dark red triangles). (**B**) Sex-specificity histograms for units recorded from estrogen-treated adult females (dark red fill), adult untreated females (red; no fill), and untreated adult males (blue; no fill) for comparison. Estrogen treatment reduced responses in adult female mice to male stimuli (leftward) and increased the frequency of responses to same-sex stimuli (rightward). Horizontal lines indicate the bootstrapped 95% confidence interval for the mean of each distribution. (**C**) Comparison of the dorsal-ventral locations of male vs predator responsive units (blue) and female vs predator responsive units (red; see *Figure 2—figure supplement 1*) in the MeA of estrogen treated females. Both male responsive and female responsive units are dorsal to predator responsive units.

where critical parameters relevant to species- and sex-specific behavioral outputs are parsed out from the complex activity patterns evoked by natural stimuli. In the discussion below, we consider in further detail, the most significant features of chemosensory processing in the MeA that emerged from our study.

## Sharpening of responses in MeA relative to AOB

To fully understand how patterned activation of vomeronasal receptors triggers specific behavioral and endocrine responses, one must determine how representations of sensory stimuli are transformed through successive stages of vomeronasal processing. Sensory cues are initially detected by VNO sensory neurons, each expressing a single VR (*Leinders-Zufall et al., 2000*; *Haga et al., 2010*; *Isogai et al., 2011*), which then project to the AOB. While some AOB units display clear selectivity for distinct VNO stimuli, AOB units typically respond to multiple sensory stimuli, and many AOB units respond to multiple stimuli associated with conflicting behavioral outcomes such as female conspecifics and predators (*Ben-Shaul et al., 2010*). Nearly all AOB output neurons project to the MeA, however, the convergence of input in the MeA lead to a different pattern of sensory responses, such that the fraction of MeA neurons responding to any given stimulus is smaller, and the specificity of MeA responses elicited by sensory stimuli is greater than observed in the AOB.

One interpretation of our data is that the low stimulus specificity observed in AOB responses represents a recording bias towards local inhibitory interneurons. This, however, seems extremely unlikely because our recordings targeted the mitral cell layer, and because ample evidence from olfactory bulb electrophysiology indicates that granule cells are virtually undetectable through extracellular recordings (*Bhalla and Bower, 1997*; *Kay and Laurent, 1999*; *Rinberg et al., 2006*;

*Ben-Shaul et al., 2010*; *Doucette et al., 2011*). Instead, the complex AOB representation we observe is likely generated by the integration of VR inputs by individual AOB mitral cells (*Wagner et al., 2006*). The nature of the units recorded from in the MeA is presently unknown, and further identification by genetic methods is an important goal for future studies.

A general feature of sensory systems is that neurons located at later stages of sensory processing are less active than those at earlier stages. Sparse coding at higher processing areas is well described in vision and olfaction (*Olshausen and Field, 1996*; *Quiroga et al., 2005*; *Poo and Isaacson, 2009*), and is thought to reflect the construction of increasingly complex sensory representations. Similarly, we find that the fraction of neurons activated by any given stimulus is approximately threefold greater in the AOB compared to the MeA. While the function of unresponsive units remains unclear, a reasonable suggestion is that these units respond to stimuli that were not presented in these experiments. Alternatively, vomeronasal-induced behaviors are strongly influenced by behavioral state (e.g.: dominant vs subordinate, estrus vs non-estrus, or adult vs juvenile), and it is possible that only a subset of MeA neurons are active in any given behavioral or physiological state.

We also found that MeA neurons of adult mice respond to vomeronasal stimuli with significantly more specificity for behaviorally distinct classes of stimuli than AOB neurons, and the number of MeA neurons responding to behaviorally conflicting stimuli, although not absent, is lower than observed in the AOB. Thus, the complex responses typical of AOB neurons are interpreted by the MeA in a manner that promotes the emergence of sensory responses more closely associated with a specific behavioral output, and may help generate unambiguous and mutually exclusive behaviors (e.g.: aggression, courtship, parenting, or predator defense). The greater MeA selectivity may result from the preferential contribution of the most selective AOB units to MeA responses, or from the transformation of AOB inputs by sensory processing within the MeA. The apparent functional disambiguation of AOB inputs by MeA units suggests a logic and specificity in AOB to MeA connectivity that strongly contrasts with the distributive pattern of projection reported from the MOB to the piriform cortex (*Sosulski et al., 2011*, *Choi et al., 2005*).

## Responses of single neurons in the MeA

As observed in the VNO and AOB, MeA responses display a relatively long latency (several seconds) and slow time-course (tens of seconds) that is likely a direct result of the slow dynamics of large non-volatile sensory stimuli that activate VRs (*Holy et al., 2000*; *Hendrickson et al., 2008*; *Ben-Shaul et al., 2010*; *Shpak et al., 2012*). Nearly half of the neurons in this study were completely silent until a preferred sensory stimulus was presented, and nearly all single-unit responses identified in this study consisted of stimulus-evoked increases in neural activity. This is consistent with *c-Fos* induction observed in large subsets of MeA neurons following exposure of animals to reproductive and defensive stimuli (*Choi et al., 2005*) and the strong monosynaptic excitatory input from AOB fibers on MeA neurons (*Bian et al., 2008*; *Niimi et al., 2012*).

Since the MeA receives a broad range of modulatory inputs from other brain regions and is likely to be affected by the animal's behavioral state, it is possible that the MeA functions differently in awake vs anesthetized animals. Sensory processing in the main olfactory system is clearly altered by anesthesia level and behavioral state (*Rinberg et al., 2006*; *Cazakoff et al., 2014*), and relating the findings from the current study to data from behaving animals will be an important step for future studies.

## Emergence of sexually dimorphic responses

The most extreme behavioral differences within a species are found in the way male vs female animals respond to the same sensory stimuli. In some species, sexually dimorphic processing of sensory cues is already prominent in the sensory epithelium. For example, male and female moths (*Manduca sexta*) and silkworms *(Bombyx mori)* express an array of olfactory receptors that are not expressed by the opposite sex (*Hansson et al., 1989*; *Nakagawa et al., 2005*; *Große-Wilde et al., 2010*). Thus, moths of either sex have access to channels of sensory information associated with sex-specific behaviors that are completely absent in the opposite sex (*Schneiderman et al., 1986*). The roundworm *C. elegans* displays a variation on this theme such that sexually dimorphic activity of primary sensory neurons (ASI) during development shapes a sexually dimorphic downstream circuit that then contributes to guiding sexually dimorphic behaviors in the adult (*White and Jorgensen, 2012*).

In other species, stimuli are similarly detected by sensory epithelia of both sexes, but are differentially interpreted by distinct circuits in the male vs female brain. For example, the *Drosophila* pheromone 11-*cis* Vaccenyl acetate (cVA) elicits sexually dimorphic behaviors, but the sensory neurons that detect cVA are anatomically and functionally indistinguishable in males and females (*Kurtovic et al., 2007*; *Datta et al., 2008*). Instead, the influence of cVA on sexually dimorphic behaviors is achieved through differential projections of cVA input to the sexually dimorphic network of *fru* expressing neurons in the central nervous system (*Datta et al., 2008*; *Ruta et al., 2010*). Even more centrally based, the capacity for male, but not female, finches to sing is attributed to circuit differences in premotor nuclei (*Nottebohm and Arnold, 1976*; *Konishi and Akutagawa, 1985*; *Kirn, 2010*). Thus, the pervasive existence of sexually dimorphic behaviors underscores their evolutionary value, but the strategies employed to achieve sexual dimorphism differ widely between species.

One of the most striking features of VNO-mediated behaviors is their sexually dimorphic expression: a male mouse elicits territorial aggression in another male mouse, but courtship in a receptive female mouse. Similarly, pups trigger infanticide in virgin males but parental behavior in virgin females. One therefore expects that sex-specific differences in the behavioral significance of certain cues be

reflected in the male- and female-specific patterns of neuronal activity within the vomeronasal pathway. However, although sexually dimorphic anatomical features have indeed long been identified at every stage of the vomeronasal pathway, including in the VNO, AOB, amygdala, and hypothalamic areas (*Guillamón and Segovia, 1997*), there is little evidence for substantial differences in sensory responses of the male and female VNO or AOB (*Nodari et al., 2008*; *Herrada and Dulac, 1997*; *Kang et al., 2009*; *Ben-Shaul et al., 2010*; *Haga et al., 2010*). Thus, downstream circuits are likely to contribute significantly to generate, or amplify, sexually dimorphic sensory processing.

Previous studies show that silencing the VNO, genetically or surgically, brings about a set of behaviors that are typically seen in the opposite sex (*Stowers et al., 2002*; *Kimchi et al., 2007*). These findings demonstrate the essential contribution of the vomeronasal system to the control of social behaviors, and show that effector circuits for behaviors of both sexes exist in the brain, and are activated or repressed by vomeronasal activity in a sex-specific manner (*Kimchi et al., 2007*). The transformation of the vomeronasal sensory representation in the MeA precisely fulfills the characteristics required of such a behavioral switch. Unlike the AOB, in which sensory responses are similar in males and females, over 80% of the recorded MeA neurons respond preferentially to stimuli from the opposite sex. Accordingly, an entire category of 'same-sex' responses is decimated in the MeA representation. Same-sex information is not entirely lost: indeed, the lack of male to male aggression in TRPC2$^{-/-}$ mice demonstrates that the representation of male stimuli, in male mice, is present in wild-type animals, depends on vomeronasal signaling, and is capable of driving robust behaviors (*Stowers et al., 2002*). Same sex-information may therefore either rely on the minimal representation found in the MeA, or be processed preferentially by other secondary vomeronasal relays.

## Developmental organization of sexually dimorphic circuits

Puberty marks the transitional period during which animals develop the secondary sex traits characteristic of adult animals, and begin to display adult-typical social behaviors. It is generally believed that testosterone accounts for most, if not all, of development of the known sexually dimorphic structures in the mammalian brain (*Morris et al., 2004*). The manner by which testosterone shapes the nervous system, however, depends on the specific neural circuit, such that cell death, synapse formation, synapse elimination, neurogenesis, and changes to neuron morphology all play important roles (*Cooke et al., 1999*; *Ibanez et al., 2001*; *Morris et al., 2004*; *Zhang et al., 2008*; *Forger & de Vries, 2010*). The influence of sex-steroids on the brain is strictly controlled according to the developmental stage, with two phases described as critical. During an initial phase, which occurs near birth, neural structures are thought to be differentially 'organized' by exposure to testosterone in males, but not in females (*Phoenix et al., 1959*; *McCarthy, 2008*). This initial phase is thought to create clear structural and anatomical differences within sexually dimorphic regions. During a second phase, which occurs during puberty, sexually dimorphic circuits are 'activated' such that they begin functioning as mature circuits capable of effecting sexually dimorphic behaviors (*Phoenix et al., 1959*).

The MeA of mice is morphologically sexually dimorphic such that males have a larger posterodorsal MeA (MeApd) than females (*Cooke et al., 1999*; *Morris et al., 2004*), a difference primarily attributed to differential cell death at birth (*Wu et al., 2009*). Our recordings from the MeA of mice in the week prior to puberty onset demonstrate that MeA units had not developed the sexually dimorphic response patterns characteristic of adult animals, although these recordings were performed after sex-specific cell death is largely complete (*Wu et al., 2009*; see ; *Forger and de Vries, 2010*). Therefore, the organizational phase of steroid-dependent MeA cell death likely provides a template for future sexually dimorphic sensory responses, but is not in itself sufficient to endow the MeA with adult function that might, for example, require further elimination of neurons with the wrong sex cue specificity. Importantly, juvenile MeA neurons are not silent but rather respond vigorously to vomeronasal stimuli in a manner very similar to AOB neurons in adult animals, demonstrating that the MeA is already an actively functioning circuit.

Taken together, our results indicate that a second phase of circuit organization is required to shape sex-specific responses in adults. Our recordings from animals with manipulated hormone levels and hormone signaling show that sexually dimorphic patterns of MeA function in adult mice are estrogen dependent. Previous experiments studying an independently made ArKO mouse line reported significant impairment in social recognition (*Bakker et al., 2002*), which may result from the failure of MeA neurons in ArKO animals to mature adult patterns of MeA responses. In some species of mammals, the influence of sex-steroids on MeA anatomy persists well into adulthood (*Cooke et al., 1999*)

supporting our conclusion that additional hormone-dependent circuit changes occur after the peri-natal organizational phase. Moreover, a multi-stage organizational model of sex-specific behavioral circuits is supported by elegant experiments in hamsters demonstrating profound rearrangements of circuits underlying social behaviors during puberty (*Zehr et al., 2006*).

Several models could account for the late development of sexually dimorphic responses observed in the MeA. Death of MeA neurons during development is important for sculpting a sexually dimorphic circuit. However, we now know that the functional properties of MeA units can be similar even though the number of cells in the male vs female MeA is different. Therefore, other processes that build on these early differences must exist, and different mechanisms for achieving proper functional sexual dimorphism in the MeA can be proposed, each providing unique and testable predictions on how neural circuits in the adult MeA develop. Synaptic input from the AOB to the MeA may be sculpted in adult mice to disproportionately favor AOB input carrying opposite sex information to the MeA. Alternatively, or in addition to, maturation of inhibition within the MeA at puberty may act to silence, or specifically shunt less robust MeA responses. This latter mechanism could act in a manner similar to the control of critical period plasticity within the cortex through a time- and steroid-dependent maturation of inhibitory connectivity (*Hensch, 2005*).

The emergence of specific topographically segregated, and sexually dimorphic MeA responses indicate that the anatomical and functional processes by which response selectivity is achieved in the MeA do not act randomly. Rather, specific circuit mechanisms yet to be uncovered must act together to allow features of vomeronasal information to be extracted from the overall AOB representation based on intrinsic MeA properties, as well as, the sex, age, and physiological status of the individual animal. Remarkably, the differences in MeA activity observed in males vs females, juveniles vs adults, or mutants vs wild-type animals correlate tightly with distinct patterns of social behaviors from each of these groups. This suggests that the age- and sex-specific transformations in MeA sensory processing uncovered by our study are likely to underlie fundamental changes in social behavior throughout development.

## Materials and methods

### Animals

Mice (adult, litters of juvenile mice, and pregnant females for estrogen treatments of newborn pups) were purchased from Charles River Laboratories (Wilmington, MA). ArKO male mice were kindly provided by Dr Evan Simpson (*Fisher et al., 1998*) and bred in house. All experiments were performed in strict compliance with the National Institute of Health and Harvard University.

### Surgical procedure for VNO stimulation

Mice were anesthetized with 100 mg/kg ketamine and 10 mg/kg xylazine and the skin overlying the throat was cut with dissecting scissors. The trachea was exposed by moving overlying gland tissue and separating the right and left sternohyoid muscles. A tracheal incision was made using fine scissors and a 15-mm length polyethylene tube (I.D. 0.76 mm, O.D. 1.22 mm, Franklin Lakes, NJ) was inserted in the caudal end of the cut trachea and held in place with Vetbond glue (3 M, St. Paul, MN). This tracheotomy was subsequently used to maintain anesthesia (0.5–2% isoflurane in pure oxygen) for the duration of the electrophysiological experiments. The rostral end of the trachea was sealed closed to prevent efflux of fluid from the cervical cavity to the nasal cavity and the VNO, or incubated and used to control airflow across the MOE. The sympathetic nerve trunk was gently isolated from connective tissue and enclosed with a stimulating cuff electrode using the carotid artery for structural support. All incisions were then closed with Vetbond, and the mouse was placed in a custom built stereotaxic apparatus. To allow for cleaning the nasal cavity and the VNO between different stimulus presentations (see below), a plastic tube was inserted below the mouth to allow suction of fluid through the naso-palatine duct. Fluid flow was regulated with a computer controlled solenoid valve (Takasago, Japan) connected to a vacuum line. Surgical silk sutures (6/0, CP Medical, Portland, OR) were inserted into the cheek skin to gently pull them laterally to prevent occlusion of the naso-palatine duct during suction.

### Surgical procedure for MOE stimulation

In experiments requiring stimulation of the MOE, the mouse was tracheotomized and delivery of volatile stimuli was achieved through computer controlled inhalation. The rostral part of the

tracheotomy incision was connected to a vacuum line through a polyethylene tube (I.D. 0.76 mm, O.D. 1.22 mm). Airflow through the nasal cavity via the tube was gated by a computer controlled solenoid valve (Takasago, Japan) and regulated by a flow controller (Omega, Stamford, CT).

## Stimuli

Urine was collected from adult female and male mice of the BALB/c, CBA (Charles River Laboratories; Wilmington, MA) or C57Bl6 (Jackson Laboratories, Bar Harbor, Maine) strains and immediately placed in liquid nitrogen, for subsequent storage in −80°C. Predator urine samples were obtained from PredatorPee (Lincoln, Maine). Male, female or predator urine samples comprised samples pooled from all three strains, or species, except when they were tested individually (*Figure 3—figure supplement 2*).

## Stimulus presentation

All sensory stimuli (olfactory and vomeronasal) were presented 5 or more times in a pseudorandomized order during each experiment. VNO Stimuli were applied by placing 1 µl of stimulus (1:100 dilution) directly into the nostril. After a delay of 20 s, a stepped square-wave stimulation train (duration: 1.6s, current: ±100 µA, frequency: 30 Hz), was applied through the sympathetic nerve cuff electrode to facilitate VNO pumping and stimulus entry to the VNO lumen. Following a second delay of 40 s, a solenoid controlling suction to the nasopalatine duct was opened and 1–2 ml of Ringer's solution was passed through the nostril and out the nasopalatine duct to cleanse the VNO. This cleansing procedure lasted 30 s and was accompanied by sympathetic trunk stimulation to facilitate stimulus elimination from the VNO lumen. Volatile stimuli were presented using computer-controlled airflow past the main olfactory epithelium and out the rostral tracheotomy. Volatile stimuli were introduced by passing the air stream through a microfiber filter (Whatman: 6823-1327) containing 30 µl of stimuli (undiluted male, female, and predator urine; 1/10 and 1/100 dilutions in mineral oil of 2-heptanone, isoamyl acetate, and acetate phenone).

## Electrophysiology

For AOB recordings, we used a 4-shank configuration with 8 recording channels per shank (NeuroNexus Technologies: a4x8-5 mm 50-200-413). A craniotomy was opened immediately rostral of the rhinal sinus, the dura was removed around the penetration site, and electrophysiological probes were advanced into the AOB at an angle of ~30° using a hydraulic micromanipulator (Siskiyou, Oregon, US). Identification of the AOB external cellular layer was performed as described in *Ben-Shaul et al. (2010)*. As previously reported for the MOB (*Bhalla and Bower, 1997*; *Kay and Laurent, 1999*; *Rinberg et al., 2006*; *Doucette et al., 2011*), large well-isolated spikes were found exclusively when the electrode tracks traversed the external cellular layer of the AOB. In contrast, negligible neural activity was observed from electrode tracks located in the granule cell layer. These results suggest that AOB projection neurons (mitral and tufted cells) represent the majority of neurons recorded in these experiments. A subset of the AOB data reported in this manuscript were originally reported in *Ben-Shaul et al. (2010)*, but have been extensively reanalyzed.

MeA recordings were made with a 2-shank configuration with 16 recording channels per shank (NeuroNexus Technologies: a2x16-10 mm 100-500-413). Briefly, a craniotomy was opened dorsal to the MeA based on stereotaxic coordinates (1.7–2.0 mm lateral from midline and 1.7–2.0 mm caudal from bregma). Electrophysiological probes were positioned using a stereotaxic manipulator (MP-285; Sutter instruments, Novato, CA) and advanced 5–6 mm from dorsal surface of the brain until a minute deflection in the electrode shaft indicated that the electrode tip had reached the ventral surface. Accurate targeting was associated with a progression of distinct electrophysiological features as the electrode probe advanced through the brain. For example, large amplitude spikes were abundant as the electrodes penetrated the hippocampus. Ventral to the hippocampus, there was a period of relative quiescence, followed by large oscillatory spikes immediately dorsal to the MeA. The spontaneous neuronal activity in the MeA was typically low and isolated action potentials were distinctly smaller in magnitude than those encountered dorsal to the MeA—we used these distinctive characteristics to improve accurate targeting during the experiment. Indeed, post-mortem histology confirmed that the majority of our recordings traversed the entire dorsoventral extent of the MeA.

## Post-mortem histological reconstruction

Electrophysiology probes were coated with one of four fluorescent dyes (DiI, DiO, DiD, Invitrogen, Carlsbad, CA; FluoroGold, Fluorochrome, LLC). At the end of each experiment, the brain was extracted,

fixed, and 50-μm coronal sections were made for histological analysis of the electrode tract location to confirm accurate stereotaxic targeting to either the AOB external cellular layer or posterior MeA (*Figure 1B*). In nearly all cases, electrode tracts were accurately targeted to either the posterior MeA or the mitral cell layer of the AOB. Data was excluded for the few recording sessions with poor targeting.

## Processing of electrophysiological data

32 recording channels were band-pass filtered (300–5000 Hz) and continuously sampled at 25 kHz using an RZ2 processor, PZ2 preamplifier, and two RA16CH head-stage amplifiers (TDT, Alachua, FL). Custom MATLAB (Mathworks, Natick, MA) programs were used to extract 3.5 ms spike waveforms from the continuous data. The number of recording channels on which a given spike was detectable depended on the inter site distance (more overlap for 50 μm, less overlap for 100 μm), and the recording target (typically more for the AOB that for the MeA).

Once a spike was detected on any given channel, waveforms were extracted for all electrode channels in close proximity (a total of 8 channels per group). Thus, each single spike was described by its voltage waveforms on 8 contiguous channels and all of these signals were used for spike sorting. Specifically, on each recording session, we calculated the 1st and 2nd principal components (PCs) for each channel. Then, each spike was described by its projections (PC loadings) on the first two PCs for each of the 8 channels in its group. Thus, the shape of each spike was described by a set of 16 PC loading values. These PC loading values were input to KlustaKwik (*Harris et al., 2000*) for automatic classification. Since the algorithm is designed to 'over cluster', clusters were then manually verified and adjusted using Klusters (*Hazan et al., 2006*). Spike clusters were evaluated by consideration of their spike shapes, projections on principal component space (calculated independently for each recording session) and autocorrelation functions.

Classification of a spike cluster as representing a 'single unit' required that the cluster displayed a distinct spike shape and was fully separated from both the origin (noise) and other clusters (multi-unit) with respect to at least one principal component projection. We also verified that the interspike interval histogram for a given 'single unit' cluster demonstrated a clear refractory period (*Figure 1—figure supplement 1*).

## Numbers of units collected for each experimental condition

A total of 3543 units were recorded from adult BALB/c mice for all experiments. 1938 units were recorded from the MeA of adult BALB/c mice of which 197 responded to VNO stimulation (106 from male animals; 91 from female animals). 1605 units were recorded from the AOB of adult BALB/c mice of which 274 responded to VNO stimulation (210 from male animals; 64 from female animals). 517 units were recorded from the MeA of juvenile BALB/c mice of which 87 responded to VNO stimulation (37 from male animals; 50 from female animals). 290 units were recorded from the MeA of adult BALB/c female mice perinatally treated with estrogen, with 48 responding to VNO stimulation. 737 units were recorded from the MeA of ArKO$^{-/-}$ and ArKO$^{+/-}$ mice of which 99 responded to VNO stimulation (41 from ArKO$^{+/-}$; 58 from ArKO$^{-/-}$).

## Statistical identification of sensory-driven responses

All statistical analyses were performed with custom Matlab code. Significant responses were identified by comparing each unit's spiking rate during the pre-stimulation (20 s prior to stimulation) and post-stimulation (40 s after stimulation) epochs using a non-parametric ANOVA performed at the significance level of $p \leq 0.01$. This post-stimulus window ensured inclusion of responses occurring prior to and after electrical stimulation of the sympathetic nerve. Response magnitude was quantified as the change in average firing rate during the 40 s following stimulus presentation relative to the firing rate during the 20 s prior to stimulus presentation. Spike rates for histograms were binned into 2 s epochs, and then averaged across repeated stimulus presentations. Unless otherwise noted, statistics comparing two populations of units (e.g.: responses from male vs female animals) were performed using a non-parametric permutation test (*Efron and Tibshirani, 1993*) as observed distributions were not typically normally distributed.

## MeA topography analysis

The precise geometry of the multichannel electrophysiology probes allowed us to determine the relative dorsal–ventral location of MeA neurons with high accuracy. First, single units with statistically significant responses to at least one stimulus were classified in a winner-take-all manner as 'conspecific

responsive' or 'predator responsive' based on the magnitude of stimulus-driven responses. Next, the physical distance between the recording sites for each pair of units (between the two categories) was calculated. For *Figure 2—figure supplement 1B*, positive distances indicate that the conspecific unit was dorsal to the predator unit. Then, we calculated a 'predator/conspecific contrast ratio', $(resp_{con} - resp_{pred})/(resp_{con} + resp_{pred})$, for each unit that ranges from −1 (entirely predator responsive) to 1 (entirely conspecific responsive). The predator/conspecific contrast ratio values were then subtracted for each predator vs conspecific unit comparison, in a manner identical to the distance analysis described above, to determine if there was a correlation between the location of a unit and the specificity of response tuning (*Figure 2—figure supplement 1C*).

## Correlation analysis

Pearson's correlation values (insets: *Figure 3C,D*; *Figure 6C*) were calculated, for all responsive MeA units, based on the array of responses to stimulus 'A' vs stimulus 'B'. To ensure that the observed correlations/regressions were not spurious results generated by a few outliers, we performed a partial leverage analysis. Analyses were then recalculated with high leverage points removed (*Figure 2—figure supplement 2*), and in no cases did the exclusion of high leverage points significantly alter the final result.

## Multidimensional scaling

A non-metric multidimensional scaling was used to reduce the dimensionality of the stimulus category distance matrix, so that the distance between each stimulus could be approximated in two dimensions based on Kruskal's normalized stress criterion (*Figure 3—figure supplement 2C*). Since this algorithm can vary when given a random seed, we performed the analysis 100 times, and the most common result (~55% of cases) was shown. No cases were observed in any of these repetitions in which the triangles formed by different categories intersected or overlapped.

## Triangular plots of neural responses

Triangle plots (*Figure 3C,D*) of sensory responses to male female and predator stimuli were generated in three steps. First, the pre-stimulus firing rate was subtracted from the post-stimulus firing rate of each unit in order to account for differences in baseline firing rates. The absolute value of the baseline subtracted response was then taken to account for responses that consisted of a reduction in activity as compared to the pre-stimulus epoch. As can be seen in *Figure 1—figure supplement 2A*, such cases are an extreme minority in our data as units typically were inactive before a stimulus was presented and respond to stimuli with an increase in activity. Second, baseline adjusted firing rates were normalized by dividing by the summed responses to all stimuli such that:

$$resp_{Male} + resp_{Female} + resp_{Pred} = 1.$$

At this point, each unit can be plotted on an equilateral triangle with vertices at (0,0,1), (0,1,0), and (1,0,0), where the three axes represent the normalized responses to male, female and predator stimuli. Third, the plane segment defined by the intersections of this plane with the three Cartesian axes was then rotated for ease of visualization.

The selectivity index (*Figure 3B*) was calculated as the normalized distance from the center of this triangle such that data with the highest selectivity (located at vertices of the triangles) had a selectivity of 1 and the least selective units (center of triangles) had a selectivity of 0. The calculation for each individual unit was as follows:

$$\text{Response selectivity} = \frac{\sqrt{(p - \overline{x})^2 + (m - \overline{x})^2 + (f - \overline{x})^2}}{d}$$

Where p, m, and f are the responses to predator, male, and female stimuli respectively. The value corresponding to equal responses to all stimuli is $\overline{x}$, and d is the maximum possible value of $\sqrt{(p - \overline{x})^2 + (m - \overline{x})^2 + (f - \overline{x})^2}$ for standardization. In our case, $\overline{x} = \frac{1}{3}$ and $d = \frac{\sqrt{2}}{\sqrt{3}}$.

## Fluoro-gold iontophoresis

During a single surgical session, a small craniotomy was made dorsal to the MeA and a glass electrode (1 mm borosilicate; 20-25 μm outer tip diameter) filled with the retrograde neuronal tracer Fluoro-Gold (Fluorochrome, LLC; 10% in water) was lowered into the MeA. Prior to insertion into the brain,

a retaining current of −2 μA was applied and the outer surface of the glass electrode was thoroughly washed, to minimize unintentional labeling along the electrode path dorsal to the injection site. At the injection site (1.80 mm lateral, 1.8 mm posterior, and 5.5 mm ventral from Bregma) the polarity of current was switched and +5 μA, constant current was applied for 10 min. Following each injection, a −2 μA retaining current was reapplied for 10 min prior to and during the retraction of the electrode. The scalp was then sutured and a small amount of Vetbond was applied to close the wound. Animals were euthanized following a 7-day post-operative survival period, and the brains were fixed in 4% paraformaldehyde in saline. Serial vibratome sections (50 μm thickness) of the entire brain and olfactory bulbs were cut, counterstained with NeuroTrace Green (Invitrogen), and mounted on gelatin-coated slides. Fluoro-Gold labeling was visualized and analyzed using a fluorescence microscope with UV filters. All histological sections containing either AOB or MOB were counted and included in the data analysis.

### Estrogen treatment

Estrogen treatments consisted of, 5 μg of 17β Estradiol-Benzoate (Sigma) dissolved in 0.1 ml sesame oil and injected subcutaneously to female pups on days P1, P8, and P15 (*Wu et al., 2009*). Mice were subsequently tested between the ages of 10 and 12 weeks.

## Acknowledgements

We thank N Uchida, V Murthy and members of the Dulac lab for their helpful critiques of this manuscript. We are particularly grateful for the many helpful conversations with A Lanjuin regarding these experiments, S Sullivan, E Soucy, and J Greenwood for technical assistance, and R Hellmiss for help preparing the figures. We also thank the reviewers at eLife for comments and suggestions that greatly improved our manuscript. JFB, YB-S, and CD conceived of these experiments, JFB and YB-S conducted the experiments and analyzed the data, JFB, YB-S, and CD wrote the paper. This work was supported by the NIH NIDCD and HHMI grants to CD and an NRSA F32 fellowship to JFB.

## Additional information

### Competing interests

CD: Senior editor, *eLife*. The other authors declare that no competing interests exist.

### Funding

| Funder | Grant reference number | Author |
| --- | --- | --- |
| National Institutes of Health | R01DC009019 | Catherine Dulac |
| National Institutes of Health | R01DC013087 | Catherine Dulac |
| Howard Hughes Medical Institute | NA | Catherine Dulac |
| National Institutes of Health | F32 DC10089 | Joseph F Bergan |

The funders had no role in study design, data collection and interpretation, or the decision to submit the work for publication.

### Author contributions

JFB, YB-S, Conception and design, Acquisition of data, Analysis and interpretation of data, Drafting or revising the article; CD, Conception and design, Analysis and interpretation of data, Drafting or revising the article

### Ethics

Animal experimentation: This study was performed within the facilities of the Harvard University Faculty of Arts and Sciences (HU/FAS) in strict accordance with the recommendations in the Guide for the Care and Use of Laboratory Animals of the National Institutes of Health. All animals were handled according to a protocol approved by the Harvard University Institutional Animal Care and Use Committee (IACUC; protocol #97-03 and #25-13). The HU/FAS animal care and use program maintains full AAALAC accreditation, is assured with OLAW (A3593-01), and is currently registered with the USDA. Every effort was made to minimize animal suffering during this study.

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
