## [Decision Letter]

Thank you for sending your work entitled “Sex-specific processing of social cues in the medial amygdala” for consideration at *eLife*. Your article has been favorably evaluated by Eve Marder (Senior editor) and 3 reviewers, one of whom, Peggy Mason, is a member of our Board of Reviewing Editors.

The Reviewing editor and the other reviewers discussed their comments before we reached this decision, and the Reviewing editor has assembled the following comments to help you prepare a revised submission.

This manuscript nicely describes refined stimulus tuning and sexual dimorphism in the MeA in the processing of non-volatile social cues detected by the vomeronasal system. This manuscript describes an elegant set of experiments investigating the response properties of individual MeA neurons to non-volatile odorant cues from predator and male and female rodent urine. They find that 1) neurons in the MeA have heightened stimulus specificity as compared with neurons in the accessory olfactory bulb, the upstream region; 2) sexually dimorphic responses that require intact estrogen signaling in the prenatal mouse brain. This is an important piece of work that should be of interest to a broad audience.

There are a large number of additional experimental details, figure adjustments, and explanatory text revisions that would enhance this paper. We look forward to a revised manuscript that addresses these points.

*Reviewer*
*#1:*

I suggest saying up front when the # of units is given that males and females were grouped together (until they're not) and how many there were of each.

I think that the response selectivity description in the Results is understated and suggest it needs defining in the Results.

Single-and multi-units were classified by spike shape, separation of principal component projections, and the presence or absence of a refractory period between successive spikes – I don't understand this for multi units.

In most experiments, ∼2-6 recording sites were located dorsal to the MeA and were easily distinguished from the quiet baseline activity of the MeA. Do you mean to say that “units located dorsal to the MeA were easily distinguished from the quiet units of the MeA”?

PLCO is not defined at first use and is not actually defined anywhere except in a figure legend. What does PLCO stand for?

In Figure 1, average histograms of the mean response (left panels) are shown with standard error (shaded region). Is this a typo and should be right panels? If not, I do not understand. Also, any SEM shading is way too small; even at 125% to see. Either increase the size of the figure or delete this.

I don't see what Figure 1—figure supplement 2 adds?

“Therefore, the dorsoventral topography is largely a result of quantitative biases in the numbers of 'conspecific' and 'predator' neurons located dorsal versus ventral, while only 3% of the variance is explained by a smooth transition from 'predator' selective neurons (ventral) to 'conspecific' selective neurons (dorsal).” This is an odd statement in the sense that no quantification is given to support the quantitative biases.

Figure 2—figure supplement 2: There are no ordinate scales. Please make the insets in Figure 3 larger in size. They are exceptionally hard to see. The authors may want to include a recent reference on awake versus anesthetized dynamics in MOB (9).

*Reviewer*
*#2:*

The experimental logic, sequence, and analyses are very good; however some additional background and description of experimental details regarding the MeA would be beneficial. For example, what is known about the functions/differences of the specific MeA nuclei? What nuclei are targeted during recordings? What types of neurons are being recorded in the MeA? Also, it would be useful to include some discussion on why only ∼10% of the neurons within the MeA are responsive to stimuli presented.

*Reviewer*
*#3*:

1) The authors could do a better job of explaining how the primary data is processed to generate the triangular plots. In the methods they say that the triangular plots are generated by “projecting the normalized male, female, and predator responses on the plane,” but do not describe what exact measure of electrical activity is used responses – change in firing rate averaged across repeats? Or explain how the responses are normalized.

2) It is not clear if the animals are exposed to just one sample of female/male/predator urine (obtained from multiple animals) or whether in each experiment they are tested with a second independent sample. This would be important to confirm that any one response is due to chemosignals from one of the three categories. There is a discussion of principal component analysis and a supplemental figure which shows that cells responsive to a female (male or predator) stimulus are generally sensitive to an independent stimulus from the same class, but the correlation appears to hover at less than 50% for male/female odors and lower for predator. Moreover, the diagram in Figure 3—figure supplement 2 seems a bit misleading, as bobcat appears closer to Male BalbC and Male C57, yet is drawn with a line connecting it, arbitrarily it seems, to rat and fox. The argument that states the predator stimuli group together seems thin, at best. The male and female stimuli are more convincingly correlated/clustered. This should be discussed.

---

## [Author Response]

Reviewer #1:

*I suggest saying up front when the # of units is given that males and females were grouped together (until they're not) and how many there were of each*.

Good point. The legend for Figure 3 now reads: “(C) Selectivity of sensory responses for units recorded in the adult AOB (197 units from male and female animals). (D) Selectivity of sensory responses for units recorded in the adult MeA (274 units from male and female animals).”

The legend for Figure 4 now reads: “(A) Responses of AOB neurons to vomeronasal stimuli in adult male (210 units) and female (64 units) mice. (B) Responses of MeA neurons to vomeronasal stimuli in adult male (106 units) and female (91 units) mice. (C) Responses of MeA neurons to vomeronasal stimuli in juvenile male (37 units) and female (50 units) mice. Units shown in panels A-C are the same data shown in Figure 3 but classified according to the sex of the animal recorded.”

*I think that the response selectivity description in the Results is understated and suggest it needs defining in the Results*.

The results now include the following statement: “To further compare the stimulus selectivity of individual neurons, we calculated a selectivity index that ranges linearly from 0 (all stimuli elicit equal responses: center of triangles in panels C,D) to 1 (only one stimulus elicits a response: vertices of triangles in panels C,D) for units recorded from the AOB (mean 0.38; 95% CI: 0.36 to 0.41; see supplementary information) and from the MeA (mean 0.54; 95% CI: 0.50 to 0.58).”

*Single-and multi-units were classified by spike shape, separation of principal component projections, and the presence or absence of a refractory period between successive spikes – I don't understand this for multi units*.

Our wording was poor; this was the process to identify single units. The text now reads: “Single units were classified from multi-unit activity based on spike shape, separation of principal component projections, and the presence or absence of a refractory period between successive spikes (Figure 1—figure supplement 1; [30]; [32]).”

*In most experiments, ∼2-6 recording sites were located dorsal to the MeA and were easily distinguished from the quiet baseline activity of the MeA. Do you mean to say that “units located dorsal to the MeA were easily*
*distinguished from the quiet units of the MeA”?*

The text has been amended as follows: “In most experiments, ∼2-6 recording sites were located dorsal to the MeA and were easily distinguished from the quiet units of the MeA in the absence of vomeronasal stimulation.”

*PLCO is not defined at first use and is not actually defined anywhere except in a figure legend. What does*
*PLCO stand for?*

PLCO is now defined properly: “MOE stimulation elicited clear olfactory-evoked neuronal activity in the MOB and the posterolateral cortical amygdala (PLCO) in response to volatile urinary cues and odorants (Figure 1—figure supplement 3).”

*In*
Figure 1*, average histograms of the mean response (left panels) are shown with standard error (shaded region). Is this a typo and should be right panels? If not, I do not understand. Also, any SEM shading is way too small; even at 125% to see. Either increase the*
*size of the figure or delete this. I don't see what*
Figure 1—figure supplement 2
*adds?*

Our wording was unclear. The legend for Figure 2 now reads: “(Left) Each row shows the responses elicited in a single MeA unit by four different VNO stimuli, with each stimulus presented 5 times. The order of stimulus presentation was randomized during the experiment, but is shown grouped by stimulus for clarity. (Right) Histograms showing the mean response and standard error (shaded region) for each unit. Responses were aligned to the onset of stimulus presentation. All significant responses (p<0.01; Nonparametric ANOVA) are indicated by an asterisk in the top right corner of the average histogram plots. Response magnitudes for each unit were normalized to the maximum response for all stimuli. Colored bars (top and bottom) indicate the 40 second epoch following stimulus presentation that was considered for all analyses.

We also agree with the second point, and Figure 1—figure supplement 2 has been removed.

*“Therefore, the dorsoventral topography is largely a result of quantitative biases in the numbers of 'conspecific' and 'predator' neurons located dorsal versus ventral, while only 3% of the variance is explained by a smooth transition from 'predator' selective neurons (ventral) to 'conspecific' selective neurons (dorsal).” This is an odd statement in the sense that no quantification is given to support the quantitative biases*.

This is a continuation of the previous regression analysis, and a direct reference to the observed R2 value of that regression: “Regression analysis indicates a weak relationship between the dorsoventral distance and the difference in stimulus specificity for unit pairs (R^2^ = 0.033, F = 8.6, p = 0.003; black line; light grey points omitted due to high leverage).”

Figure 2—figure supplement 2*: There are no ordinate scales. Please make the insets in*
Figure 3
*larger in size. They are exceptionally hard to see. The authors may want to include a recent reference on awake versus anesthetized dynamics in MOB (*[9]*)*.

Both figures were amended as suggested and the [9] reference has been added: “Sensory processing in the main olfactory system is clearly altered by anesthesia level and behavioral state (Rinberg et al., 2006; [9])…”

Reviewer #2:

*The experimental logic, sequence, and analyses are very good; however some additional background and description of experimental details regarding the MeA would be beneficial. For example, what is known about the functions/differences of the specific MeA nuclei? What nuclei are targeted during recordings? What types of neurons are being recorded in the MeA? Also, it would be useful to include some discussion on why only ∼10% of the neurons within the MeA are responsive to stimuli presented*.

We have addressed each of these questions:

*For example, what is known about the functions/differences of the specific MeA nuclei? What nuclei are*
*targeted during recordings?*

One of the best-known functional features of the MeA, demonstrated via immediate early gene studies, is the segregation into dorsal and ventral processing subdivisions. The posterior dorsal MeA, or MeApd, is essential for reproductive and social behaviors while the ventral MeA, or MeApv, is essential for defensive behaviors (Fernandez-Fewell et al., 1994; [11]; [94]). We stereotaxically targeted the posterior MeA such that the array of recording sites would span both the MeApv and MeApd.

*What types of neurons are being recorded*
*in the MeA?*

While one can make reasonable predictions regarding which neurons likely constitute our MeA recordings, we simply don't know the answer to this question yet. We now directly address this issue in the text: *“*The nature of the units recorded from in the MeA is presently unknown, and further identification by genetic methods is an important goal for future studies.”

*Also, it would be useful to include some discussion on why only ∼10% of the neurons within the MeA are responsive to stimuli presented*.

We now discuss this issue more directly. A general feature of sensory systems is that neurons located at later stages of sensory processing are less active than those at earlier stages. Sparse coding at higher processing areas is well described in vision and olfaction (72; 76; 78), and is thought to reflect the construction of increasingly complex sensory representations. Similarly, we find that the fraction of neurons activated by any given stimulus is approximately 3 fold greater in the AOB compared to the MeA. While the function of unresponsive units remains unclear, a reasonable suggestion is that these units respond to stimuli that were not presented in these experiments. Alternatively, vomeronasal induced behaviors are strongly influenced by behavioral state (for example: dominant versus subordinate, estrus versus non-estrus, or adult versus juvenile), and it is possible that only a subset of MeA neurons are active in any given behavioral or physiological state.

Reviewer #3:

*1) The authors could do a better job of explaining how the primary data is processed to generate the triangular plots. In the methods they say that the triangular plots are generated by “projecting the normalized male, female, and predator responses on the plane,” but do not describe what exact measure of electrical activity is used responses – change in firing rate averaged across repeats? Or explain how the responses are normalized*.

We have expanded and clarified this description:

“Triangular plots of neural responses: Triangle plots (see Figure 3) of sensory responses to male female and predator stimuli were generated in three steps. First, the pre-stimulus firing rate was subtracted from the post-stimulus firing rate of each unit in order to account for differences in baseline firing rates. The absolute value of the baseline subtracted response was then taken to account for responses that consisted of a reduction in activity as compared to the pre-stimulus epoch. As can be seen in Figure 1—figure supplement 2, such cases are an extreme minority in our data as units typically were inactive before a stimulus was presented and respond to stimuli with an increase in activity. Second, baseline adjusted firing rates were normalized by dividing by the summed responses to all stimuli such that:

resp_Male_ + resp_Female_ + resp_Pred_ = 1.

At this point, each unit can be plotted on an equilateral triangle with vertices at (0,0,1), (0,1,0), and (1,0,0) where the three axes represent the normalized responses to male, female and predator stimuli. Third, the plane segment defined by the intersections of this plane with the three Cartesian axes was then rotated for ease of visualization.”

*2) It is not clear if the animals are exposed to just one sample of female/male/predator urine (obtained from multiple animals) or whether in each experiment they are tested with a second independent sample. This would be important to confirm that any one response is due to chemosignals from one of the three categories. There is a discussion of principal component analysis and a supplemental figure which shows that cells responsive to a female (male or predator) stimulus are generally sensitive to an independent stimulus from the same class, but the correlation appears to hover at less than 50% for male/female odors and lower for predator. Moreover, the diagram in*
Figure 3—figure supplement 2*seems a bit misleading, as bobcat appears closer to Male BalbC and Male C57, yet is drawn with a line connecting it, arbitrarily it seems, to rat and fox. The argument that states the predator stimuli group together seems thin, at best. The male and female stimuli are more convincingly correlated/clustered. This should be discussed*.

The stimuli presented are now clarified in the results section: “Unless otherwise noted, predator stimuli were a mixture of bobcat, fox, and rat urine diluted 1:100 in Ringer's solution; female and male stimuli were a mixture of urine from balbC, C57, and CBA strains diluted 1:100 in Ringer's solution.”

During the categorization experiments (Figure 3—figure supplement 2), 1:100 diluted unmixed stimuli were presented. The reviewer is correct that the correlations between stimuli from the same class are clear, however, there is clear diversity in responses across these experimenter-defined categories. This diversity is most pronounced for the predator category.

Regarding the lines connecting each stimulus in panel C, these lines represent categories defined a priori and are not the result of an unbiased categorization. This is now clarified in the figure legend: “Multidimensional scaling analysis shows that stimuli from a single category populate a region of the response space that is non–overlapping with stimuli drawn from a different category. Lines connecting individual data points indicate ethologically defined stimulus categories.”

*3) Which predator urine was used for the main series of experiments? This should be mentioned in the text*.

This is now defined (see above).